

# Determining the hip joint isokinetic muscle strength and range of motion of professional soccer players based on their field position

Ali AlTaweel, Shibili Nuhmani, Mohammad Ahsan, Turki Abualait and Qassim Muaidi

Department of Physical Therapy, College of Applied Medical Sciences, Imam Abdulrahman Bin Faisal University, Dammam, Saudi Arabia

## ABSTRACT

**Background**. Soccer players' physical and physiological demands vary based on their field position. Although the hip joint has an important role in soccer, little information is available about the strength and flexibility of the hip joint based on player positions. Therefore, this study aims to investigate the differences in muscle strength and flexibility of the hip joint of professional soccer players based on their field position.

**Methods**. Ninety-six professional soccer players from Saudi Arabia were divided into four groups (goalkeepers, defenders, midfielders, and attackers), with 24 participants in each group based on their field position. The Modified Thomas test was used to measure the hip extension range of motion (ROM), and muscle strength was assessed by an Isokinetic dynamometer.

**Results**. There were no statistically significant differences in the isokinetic strength at the hip joint movements between goalkeepers, defenders, midfielders, and attackers ($p \geq 0.05$). At the same time, there was a significant difference between groups in the hip extension ROM ($p \leq 0.05$). according to different player positions. *Post hoc* tests reported significant differences between goalkeepers and defenders ($p \leq 0.05$), midfielders ($p \leq 0.05$), and attackers $p \leq 0.05$). At the same time, there were no significant differences between defenders and midfielders ($p \geq 0.05$), defenders and attackers ($p \geq 0.05$), and midfielders and attackers ($p \geq 0.05$).

**Conclusion**. Even though there was no significant difference in isokinetic strength, there was a significant difference in hip extension ROM among players based on field position. This study may help coaches and trainers to recognize the strengths and weaknesses of players and design training programs to rectify the weaker components and improve players' performance in different playing positions.

## INTRODUCTION

Soccer has unique characteristics with various motor skills compared to other sports. A study investigating the physical demands of soccer players showed that soccer contains various activities such as repetitive short duration sprints, high-intensity running, walking,

Corresponding author
Shibili Nuhmani,
snuhmani@iau.edu.sa

standing, jumping, tackling, changing directions, and backwards running (*Saeidi & Khodamoradi, 2017*). Furthermore, these activities could take a high intensity that reaches around 80 to 90% of the maximal heart rate (*Saeidi & Khodamoradi, 2017*). Soccer coaches and players must consider the variables that can affect the player's performance to achieve optimal performance levels. Thus, for the player to be capable of the match physical demands, they must have physiological characteristics such as aerobic and anaerobic fitness and have good muscle strength in the upper and the lower limb (*Rathinakamalan & Senthilvelan, 2011*).

The performance during the match may differ depending on many factors such as the level of the player, the level and the demands of the match, the style, and the tactics of the game (*Ruas et al., 2015*; *Tourny-Chollet, Leroy & Beuret-Blanquart, 2000*). A previous study reported that the central midfield player covered a high average of around 12 km while the defender covered the smallest distance with 10.6 km, and the striker is between the two players' averages, with 11.25 km (*Di Salvo et al., 2007*). The differences between players in different positions appeared in the activities they perform, such as running, slide tackling, heading the ball, and long passing, which put an extra physiological load on the players (*Ruas et al., 2015*) in their different positions. Differences between players regarding their age, body mass index, and stature have been reported among soccer players of different positions, suggesting that some player's characteristics, such as body shape and size, may be appropriate to the demands of the playing position (*Bloomfield et al., 2005*). Not only the body characteristics but the position role has its influence on the players. It affects the total energy expenditure during the match, suggesting that the different play positions require different physiological, bioenergetic, and physical demands (*Di Salvo & Pigozzi, 1998*; *Reilly, 1997*). Physical and physiological demands have been varied among soccer players based on their field position. During soccer matches, midfielders cover around 5–15% more distance than defenders and attackers and even 20–40% more distance with high intensity than attackers (*Rampinini et al., 2009*). Attackers and fullbacks spend 20–40% more time sprinting than other players (*Stølen et al., 2005*). These studies show that each playing position has specific demands and activities in soccer.

Muscle strength and flexibility have a significant role in soccer performance. Isokinetic peak torque has been used to assess muscle strength in the lower limbs among soccer players (*Daneshjoo et al., 2013*; *Goulart, Dias & Altimari, 2007*; *Silva et al., 2015*). Previous studies reported a difference in muscle strength among soccer players based on their position on the field (*Goulart, Dias & Altimari, 2007*; *Lehance et al., 2009*; *Öberg et al., 1986*; *Ruas et al., 2015*; *Tourny-Chollet, Leroy & Beuret-Blanquart, 2000*). *Öberg et al. (1986)* found differences in concentric isokinetic peak torque of the hamstring and quadriceps muscles between the lowest and the highest level Swedish soccer player. *Tourny-Chollet, Leroy & Beuret-Blanquart (2000)* revealed that the forward players had significantly higher hamstring concentric strengths than midfielders and defenders. *Ruas et al. (2015)*demonstrated that goalkeepers showed different characteristics and isokinetic concentric strength across muscles than other players on the soccer field. *Goulart, Dias & Altimari (2007)* indicated that fullback players showed lower flexor muscle strength than the other soccer players, and goalkeepers' extensor muscle strength showed lower

isokinetic strength than other players. *Lehance et al. (2009)* also demonstrated that the isokinetic muscle strength of flexors and extensors increases with the sporting level and the age of soccer players.

Moreover, soccer players need to have good flexibility in muscles and joints. Reduced flexibility can risk the player and get injured during training and matches. High levels of flexibility is necessary to reach the best performance during sports participation (*Chu & Vermeil, 1983*; *Herbert & Gabriel, 2002*). The range of motion (ROM) of the hip joint has a key role in soccer performance. It is one of the most important lower limb joints in soccer-related activities such as kicking and sprinting (*Light, 2018*). According to *Amiri-Khorasani, Osman & Yusof (2011)*, a higher ROM of the hip joint positively impacts the angular velocity of the lower limb during kicking activities during forward and follow-through phases. Reduced hip ROM is also a risk factor for various hip-related injuries among soccer players (*Ibrahim, Murrell & Knapman, 2007*). Injury prevention requires a balance of muscular strength and flexibility between the anterior and posterior muscle groups of the thighs (*Daneshjoo et al., 2013*)

Even though there are several studies available in the literature on the isokinetic muscle strength of soccer players, little information is available about hip muscle strength and flexibility profiles in terms of player positions (*Gorostiaga et al., 2009*; *Śliwowski et al., 2017*). Investigating the differences in the hip muscle strength and flexibility according to soccer players' field position may help the coaches, trainers, and sports medical teams design appropriate training programs and prevent injury based on their position on the field. It may also provide information about the factors that reduce the player's performance and the physical characteristics critical to success in that position. So, this study aims to investigate the differences in muscle strength and flexibility of the hip joint of professional soccer players based on their field position. We hypothesized that there are significant differences in isokinetic hip muscle strength and hip flexibility in professional soccer players based on their field position.

## METHODS AND MATERIALS

### Participants

The volunteer participants consist of 96 professional male soccer players with mean age: of 23.11 ± 8.7 years; height: of 174.92 ± 6.27 cm; body weight: of 69.99 ± 8.68 kg; body mass index: 22.84 ± 2.29, from various first division soccer clubs in Saudi Arabia. The participants were divided into four groups (*i.e.*, goalkeepers, defenders, midfielders, and attackers), with 24 participants in each group based on their field position. The sample size was calculated as 24 in each group to estimate a difference of 4.9 N.m in the isokinetic strength of hip extensors based on a previous study (*Masuda et al., 2005*) with 80% power and 5% significance level. The players were invited through announcement flyers distributed to all who were enrolled in Saudi soccer clubs, which were part of the Saudi professional league, Saudi league 1 division, and Saudi Olympic League. Players who volunteered have been checked for participation eligibility according to inclusion and exclusion criteria. All the participants had more than four years of experience in competitive

soccer events. Participants with a history of musculoskeletal injury in the lower limb and back in the past three months, any neurological problem, systemic diseases and use of any medication or any biomechanical abnormalities that affect the performance were excluded from the study. A cross-sectional study design was adopted to conduct this study. This study was conducted in the eastern province of Saudi Arabia. Ethical approval of the study was obtained from the Institute Review Board of Imam Abdulrahman Bin Faisal University (IRB-PGS-2018-03-011). All the participants took part in this study voluntarily and written informed consent was obtained from them prior to participation.

## Outcome measurements
### Isokinetic muscle strength testing
A Biodex multi-joint system isokinetic dynamometer (Biodex Medical Systems, Shirley, New York, USA) was used to measure the isokinetic peak torque of the hip joint muscles. The muscle tested were Hip Flexors, extensors, abductors and adductors. Biodex multi-joint system is reliable and valid for measuring hip muscle strength at different angles for training or testing (Supine position-hip extension: ICC = 0.90; 95% CI [0.85–0.96]; hip flexion: ICC = 0.72; 95% CI [0.46–0.99]) (*Contreras-Díaz et al., 2021*).

## Modified Thomas test
The modified Thomas test was used to measure the hip extension ROM to determine the differences between the players. *Cady, Powis & Hopgood (2022)* demonstrated that the modified Thomas test has good inter-rater reliability and intra rater reliability for the measurement of flexibility of hip joint muscles (For iliopsoas Cronbach's alpha $C\alpha = 0.95$, Fleiss kappa $F\kappa = 0.78$; For rectus femoris Cronbach's alpha $C\alpha = 1.00$, Fleiss kappa $F\kappa = 0.80$)

## Procedure
All the participants were screened in the physiotherapy lab at Imam Abdulrahman bin Faisal University based on inclusion and exclusion criteria to ensure that they were fit for the study. Ten minutes of explanation about the research were provided to the subjects before the commencement of the study. The anthropometric data of players were collected, including age, weight, height, and body mass index (BMI) was calculated. The participants filled out a paper has their names, ages, the club they play, position, and years of participation in soccer. After all of the participant's information was collected, the participant got ready to perform the modified Thomas test and then the hip isokinetic muscle measurement. The dominated limb was tested, which was determined by asking the participant about his preferred kicking leg.

To perform modified Thomas test, the participant sits on the edge of the examiner table with the ischial tuberosity clear to the table's edge. The participants rest their feet flat on the ground and then lay supine. If the preferred leg was the right leg, the participants flexed the left lower limb at the hip and knee, bringing the knee to the chest. While the participants were grasping their left knee to the chest, they rolled back into the table with assistance from the examiner. A small pillow was put under the participant's lumbar region to avoid the lumbopelvic movement (*Kim & Ha, 2015*; *Vigotsky et al., 2016*). The right hip

is freely extended to allow full hip extension without support from the examiner's table until the participant feels completely relaxed. A universal goniometer measured the hip extension angle with the fulcrum placed over the lateral aspect of the greater trochanter, the proximal arm aligned on the lateral midline of the pelvis and the distal arm aligned laterally to the femur using the lateral epicondyle as a reference point (*Clapis, Davis & Davis, 2008*; *Wakefield et al., 2015*; *Vigotsky et al., 2016*) (Fig. 1).

For the isokinetic strength testing, the participants performed ten minutes warm-up by a cycle ergometer without resistance. The participants performed the maximal contraction through the ROM. They performed the isokinetic concentric contraction measuring the maximal strength of the hip muscles at an angular velocity of 90o/s and 180o/s. The velocity range (a slow [90o/s] and a rapid [180o/s]) concentric-mode test was selected for high reproducibility regardless of the body side, test velocity and contraction mode (*Al-Harbi, Muaidi & Ahsan, 2021*). To measure the hip flexor and extensor strength, the participant was in the supine position, and the hip attachment of the isokinetic dynamometer was inserted into the knee adaptor and secured to the dynamometer. The participant was positioned in the supine position lateral to the dynamometer system, the limb in the neutral position, with the dynamometer's axis aligned superior and anterior to the greater trochanter. The test contains two sets, each consisting of five repetitions with a rest period of 60 s between the sets. The participants were in the side-lying position to perform isokinetic testing of hip abduction and adduction. The outcome parameter was the peak torque (expressed in Nm) which was normalized to the body weight (expressed in Nm/kg) (*Timmons, Claiborne & Pincivero, 2003*). The highest peak torque value was recorded and used for statistical analysis. A rest period of 5 min was given between each muscle group's tests.

### Statistics analysis

All the statistical analyses were done by IBM SPSS program software version 21.0. on Windows. Descriptive and interferential statistics were provided by means and standard deviations. Normality and homogeneity of the data were confirmed using Shapiro Wilk's test and Leven's test ($p > 0.05$). One-way ANOVA was used to determine the differences between all variables, including the hip strength and flexibility according to the players' position. Multiple comparisons were made by using the Tukey *post hoc* test. In addition, partial eta squared ($\eta p2$) was used to report effect size. The significance level was set at $P < .05$ and confidence interval 95%.

## RESULTS

The result showed a significant difference in weight ($p = .001$), height ($p = .008$), and BMI ($p = .043$) between playing positions, as shown in Table 1. The goalkeepers were significantly heavier ($178.42 \pm 5.66$ kg) than the other groups ($p < 0.05$). The goalkeepers were significantly taller ($175.21 \pm 7.23$) than defenders and midfielders ($p = .005$ and $p = .002$), respectively. In the BMI, the result shows that the goalkeepers were significantly higher ($24.02 \pm 2.98$ kg/m2) than all-out field positions ($p < 0.05$).

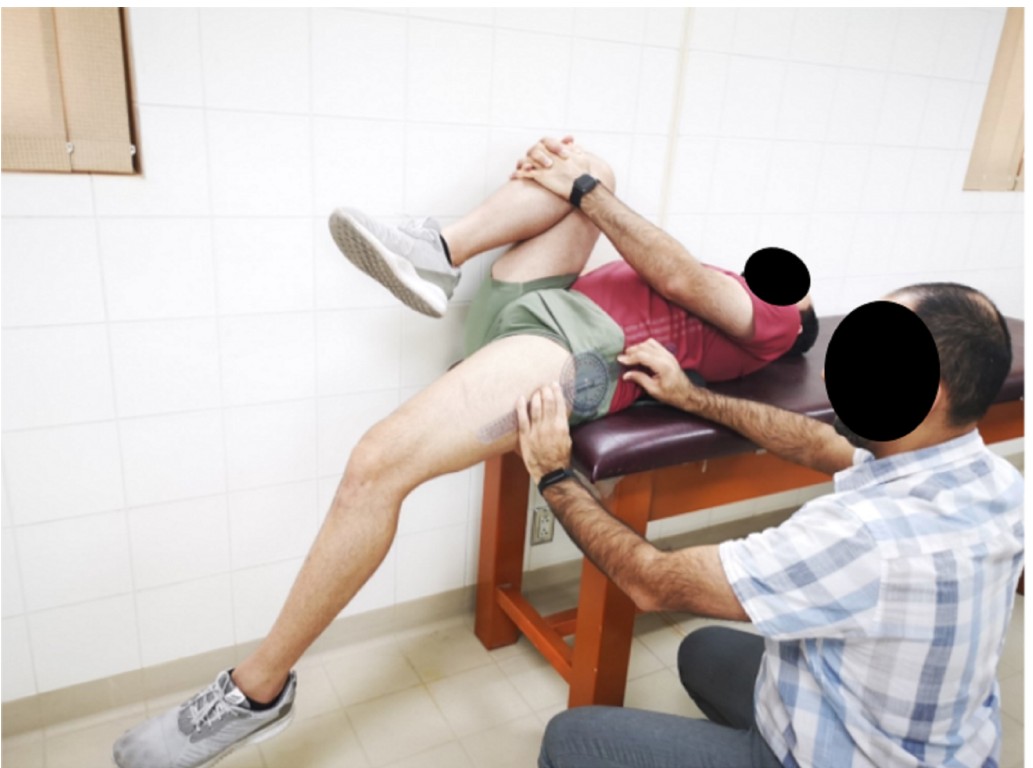

**Figure 1** Modified Thomas test assessment of hip extension ROM to assess hip flexor muscle flexibility.

There were no statistically significant differences in the isokinetic strength at the hip joint movements between goalkeepers, defenders, midfielders, and attackers ($p > 0.05$) (Table 2). At the same time, there was a significant difference between groups in the hip extension ROM ($p = .000$, $\eta p2 = .475$) according to different player positions (Table 3). Further, the Post Hoc test reported significant differences between goalkeepers and defenders ($p = .000$), midfielders ($p = .000$), and attackers ($p = .000$) in hip extension ROM. At the same time, there were no significant differences between defenders and midfielders ($p > 0.05$), defenders and attackers ($p > 0.05$), and midfielders and attackers ($p > 0.05$) (Table 4).

## DISCUSSION

The purpose of the study was to investigate the differences in hip isokinetic strength and hip joint flexibility among professional soccer players based on their field positions. This study showed no significant differences among players in all different playing positions in hip isokinetic muscle strength. The goalkeepers had significantly higher hip extension ROM than all other players.

Among the wide range of studies on the isokinetic muscle strength of soccer players, little information is available in the literature about hip muscle strength profiles in terms of player positions (*Gorostiaga et al., 2009*; *Śliwowski et al., 2017*). In this study, there was

**Table 1  Anthropometric characteristics of the elite soccer players according to different player positions participants.**

| Variables | Positions | N | Mean ± SD | 95% CI Lower–Upper limit | Sig. |
|---|---|---|---|---|---|
| Age (years) | Goalkeepers | 24 | 23.58 ± 3.69 | 22.02–25.14 | |
| | Defenders | 24 | 22.63 ± 3.57 | 21.12–24.13 | |
| | Midfielders | 24 | 23.08 ± 5.18 | 20.89–25.27 | 0.904 |
| | Attackers | 24 | 23.13 ± 4.96 | 21.03–25.22 | |
| Weight (kg) | Goalkeepers | 24 | 76.79 ± 12.38 | 71.57–82.02 | |
| | Defenders | 24 | 67.82 ± 7.38 | 64.70–70.93 | |
| | Midfielders | 24 | 66.68 ± 6.04 | 64.13–69.23 | 0.001* |
| | Attackers | 24 | 68.70 ± 8.91 | 64.94–72.46 | |
| Height (cm) | Goalkeepers | 24 | 178.42 ± 5.66 | 176.03–180.81 | |
| | Defenders | 24 | 173.13 ± 7.11 | 170.12–176.13 | |
| | Midfielders | 24 | 172.63 ± 5.06 | 170.49–174.76 | 0.008* |
| | Attackers | 24 | 175.21 ± 7.23 | 172.16–178.26 | |
| BMI (kg/m2) | Goalkeepers | 24 | 24.02 ± 2.98 | 22.76–25.28 | |
| | Defenders | 24 | 22.65 ± 2.32 | 21.67–23.63 | |
| | Midfielders | 24 | 22.38 ± 1.83 | 21.60–23.15 | 0.043* |
| | Attackers | 24 | 22.32 ± 2.04 | 21.46–23.18 | |

no significant difference in hip muscle strength between the players according to the player positions in all directions with different speeds. Our result disagrees with another study that found the defenders and midfielders were significantly greater than goalkeepers and strikers in eccentric hip abduction and adduction strength (*Wik, Auliffe & Read, 2018*). Our study found no differences in hip muscle strength in any playing positions. Furthermore, the similarity in hip strength between all positions may be due to strength training similarity for all players, which explains the lack of differences between the players in specific positions. In our study, the value of hip muscle strength was in a lower range normally reported in the literature for elite soccer players.

The possible explanation of differences in hip extension ROM among the players based on their playing positions, in which goalkeepers are significantly higher in hip muscle flexibility could be position-specific demands. In soccer, it is clear that each position requires the player to perform specific tactics and responsibilities according to the playing positions, which may lead to similarities between the outfielders, such as the muscle strength between positions (*Magalhães et al., 2001*). Additionally, that might be because of the myriad of activities performed by outfielders during training and game (*Öberg et al., 1986*; *Magalhães et al., 2001*). Goalkeepers' responsibilities are performed differently from those of the outfield players. Outfielders' position requires covering longer distances while goalkeepers' do not (*Sales et al., 2014*).

The goalkeeper's position has a specific nature that might be completely different due to the responsibilities and performance during training and games. Goalkeeper activities rely on explosive lateral movements, dives, and jumps (*Lees & Nolan, 1998*; *Eirale et al, 2014*).

**Table 2 Isokinetic strength values of soccer players for each playing position.**

| Isokinetic strength | | Positions | N | Mean ± SD | 95% CI | Partial Eta Squared ($\eta p2$) | Sig. |
|---|---|---|---|---|---|---|---|
| Speed | Movement | | | | Lower–Upper limit | | |
| 90 | FLX | Goalkeepers | 24 | 100.68 ± 20.39 | 92.07–109.29 | .024 | .514 |
| | | Defenders | 24 | 95.97 ± 15.61 | 89.38–102.56 | | |
| | | Midfielders | 24 | 92.33 ± 19.27 | 84.20–100.47 | | |
| | | Attackers | 24 | 94.73 ± 22.49 | 85.23–104.23 | | |
| | EXT | Goalkeepers | 24 | 155.13 ± 45.71 | 135.83–174.43 | .051 | .181 |
| | | Defenders | 24 | 153.75 ± 38.16 | 137.64–169.86 | | |
| | | Midfielders | 24 | 132.06 ± 40.94 | 114.78–149.35 | | |
| | | Attackers | 24 | 146.68 ± 35.11 | 131.85–161.50 | | |
| 180 | FLX | Goalkeepers | 24 | 88.92 ± 20.09 | 80.43–97.40 | .034 | .358 |
| | | Defenders | 24 | 85.91 ± 14.11 | 79.95–91.87 | | |
| | | Midfielders | 24 | 79.99 ± 17.21 | 72.72–87.26 | | |
| | | Attackers | 24 | 84.43 ± 17.88 | 76.88–91.98 | | |
| | EXT | Goalkeepers | 24 | 133.54 ± 47.13 | 113.64–153.44 | .063 | .110 |
| | | Defenders | 24 | 134.09 ± 38.89 | 117.67–150.51 | | |
| | | Midfielders | 24 | 107.20 ± 50.16 | 86.02–128.39 | | |
| | | Attackers | 24 | 128.60 ± 34.90 | 113.87–143.34 | | |
| 90 | ABD | Goalkeepers | 24 | 63.73 ± 18.91 | 55.75–71.71 | .019 | .753 |
| | | Defenders | 24 | 61.75 ± 18.96 | 58.98–70.60 | | |
| | | Midfielders | 24 | 63.44 ± 13.74 | 57.64–69.24 | | |
| | | Attackers | 24 | 67.12 ± 17.59 | 59.69–74.55 | | |
| | ADD | Goalkeepers | 24 | 92.20 ± 46.41 | 72.60–111.80 | .013 | .741 |
| | | Defenders | 24 | 94.52 ± 34.98 | 79.75–109.29 | | |
| | | Midfielders | 24 | 87.09 ± 40.74 | 69.89–104.29 | | |
| | | Attackers | 24 | 99.70 ± 35.36 | 84.77–114.63 | | |
| 180 | ABD | Goalkeepers | 24 | 41.78 ± 21.56 | 32.68–50.88 | .020 | .609 |
| | | Defenders | 24 | 37.54 ± 13.76 | 31.73–43.35 | | |
| | | Midfielders | 24 | 36.65 ± 14.89 | 30.36–42.94 | | |
| | | Attackers | 24 | 41.78 ± 17.16 | 34.53–49.03 | | |
| | ADD | Goalkeepers | 24 | 81.13 ± 44.85 | 62.19–100.07 | .034 | .357 |
| | | Defenders | 24 | 72.32 ± 36.23 | 57.02–87.61 | | |
| | | Midfielders | 24 | 62.55 ± 39.17 | 46.00–79.09 | | |
| | | Attackers | 24 | 78.17 ± 32.24 | 64.55–91.78 | | |

**Notes.**
90, Concentric 90 deg/s; 180, Concentric 180 deg/s; FLX, Flexion; EXT, Extension; ABD, Abduction; ADD, Adduction.

The players of different positions, such as defenders, midfielders, and attackers, receive the same training and tactical training as the coaches during the daily training sessions, but the goalkeepers receive more specific positional exercises and training. Goalkeepers also have individual goalkeeper coaches who design physical and technical training programs that fit the position requirements. They are more flexible, as they usually perform to defend their goals by stretching their bodies in front of the goal area. Goalkeepers also receive specific
**Table 3**  Results of the hip ROM of the elite soccer players according to different player positions participants in the current study.

| Variables | Positions | N | Mean ± SD | 95% CI Lower–Upper limit | Partial Eta Squared ($\eta p2$) | Sig. |
|---|---|---|---|---|---|---|
| HIP ROM | Goalkeepers | 24 | 19.79 ± 1.82 | 19.02–20.56 | .475 | .000* |
| | Defenders | 24 | 16.63 ± 2.34 | 15.64–17.61 | | |
| | Midfielders | 24 | 16.71 ± 1.81 | 15.95–17.47 | | |
| | Attackers | 24 | 16.00 ± 3.45 | 14.54–17.46 | | |

Notes.
*Significant at .05 level.
ROM, range of motion.

**Table 4**  The results of *post hoc* (Tukey) test of hip ROM for soccer players according to different playing positions.

| Dependent variable | (I) GROUP | (J) GROUP | Mean difference (I–J) | Std. error | Sig. |
|---|---|---|---|---|---|
| HIP ROM | Goalkeepers | Defenders | 3.17* | 0.71 | .000* |
| | | Midfielders | 3.08* | 0.71 | .000* |
| | | Attackers | 3.79* | 0.71 | .000* |
| | Defenders | Goalkeeper | −3.17* | 0.71 | .000* |
| | | Midfielders | −.08 | 0.71 | .999 |
| | | Attackers | .63 | 0.71 | .813 |
| | Midfielders | Goalkeeper | −3.08* | 0.71 | .000* |
| | | Defenders | .083 | 0.71 | .999 |
| | | Attackers | .71 | 0.71 | .748 |
| | Attackers | Goalkeepers | −3.79* | 0.71 | .000* |
| | | Defenders | −.62 | 0.71 | .813 |
| | | Midfielders | −.71 | 0.71 | .748 |

Notes.
*Significant at .05 level.
ROM, range of motion.

training that might improve their skills, making them more flexible than the other players (*Deprez et al., 2015*).

The present study has its limitations. First, the measurement was taken from the dominant leg of the players, while the non-dominant leg was not included in the analysis. Second, the analysis did not include other factors such as hours of training, history, playing minutes, and playing tactics. Third, the study did not investigate those players who may play in different positions due to a long time of a player absence which may have influenced the results. Fourth, our study players are divided into four: goalkeepers, defenders, midfielders, and attackers. In contrast, other studies divided the players into more than four positions, such as the fullback, and external midfielders, which gives more information about the players in different positions (*Buchheit et al., 2010*; *Lago-Peñas et al., 2011*; *Markovic & Mikulic, 2011*; *Mendez-Villanueva et al., 2013*). Finally, the study did not include female soccer players, which cannot be generalized to both gender players.

## CONCLUSION

Our study showed no significant difference in hip muscle strength between the groups. There were differences in players' hip extension ROM according to the playing position. It showed that the goalkeeper was more flexible and significantly higher than midfielders. The study may help coaches recognize their players' strengths and weaknesses and design training programs to improve the weaker components and improve players' performance in different playing positions and design appropriate training programs for individuals and groups of players based on the players' position.

### Funding

The authors received no funding for this work.

### Competing Interests

Shibili Nuhmani is an Academic Editor for PeerJ.

### Author Contributions

- Ali AlTaweel conceived and designed the experiments, performed the experiments, prepared figures and/or tables, and approved the final draft.
- Shibili Nuhmani conceived and designed the experiments, authored or reviewed drafts of the article, and approved the final draft.
- Mohammad Ahsan performed the experiments, analyzed the data, authored or reviewed drafts of the article, and approved the final draft.
- Turki Abualait performed the experiments, prepared figures and/or tables, authored or reviewed drafts of the article, and approved the final draft.
- Qassim Muaidi analyzed the data, prepared figures and/or tables, and approved the final draft.

### Human Ethics

The following information was supplied relating to ethical approvals (i.e., approving body and any reference numbers):

Imam Abdulrahman Bin Faisal University

### Data Availability

The raw data is available in the Supplementary File.

### Supplemental Information

Supplemental information for this article can be found online at http://dx.doi.org/10.7717/peerj.14000#supplemental-information.

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
