# Peer review of "Determining the hip joint isokinetic muscle strength and range of motion of professional soccer players based on their field position"

_PeerJ, doi:10.7717/peerj.14000_

## Round 0.1 · original submission · Major Revisions

Dear authors,

The manuscript is very interesting. Please follow the reviewers' suggestions carefully and respond to their comments point by point.

Thanks.

·

Basic reporting

- Line 58-60 and line 60-63 contain the same information.
- Line 60-63 Please add the reference of this study.
- Line 71-72 As above, please add a reference for this statement.
- Line 73-74 Again, what are these studies? Please add the relevant references.
- Since from line 71 to line 83 argue the strength of soccer players, and from line 83 authors introduce the flexibility, I suggest beginning this part of the background with “moreover…”.
- Line 85 “…better quality of life…”. This concept is off topic. Authors’ paper focus on performance in soccer. This is a sport science study and not a public health study. Please follow the study topic.
- Although authors provide a sufficient background on the importance of strength in soccer players, they failed to argue the other characteristic they investigated, that is the flexibility. For example, I suggest considering this article "Light N. (2019). The effect of acute match play loading on hip adductor strength and flexibility in soccer players. The Journal of sports medicine and physical fitness, 59(2), 325–329." Authors should provide a background with relevant literature on this.
- The introduction should be improved. In fact, it begins by addressing the physical and physiological demands of soccer players, then it focused on strength and flexibility, and finally the authors returned to physical and physiological demands. The reading is redundant, please review.
- Furthermore, the introduction is at times too general. For example, authors referred to differences between players regarding age, BMI, or player level. In this paper, the authors did not explore differences in strength and flexibility considering weight, height, or BMI of the participants. And again, why authors reported physiological differences such as aerobic and anaerobic fitness in relation to players’ field position? Please make the introduction more specific.
- The English language should be improved.

Experimental design

- After reported the aim of the study, authors should provide their hypotheses.
- Did the authors consider the dominance of the lower limb as an inclusion criterion?
- Authors reported that for the Modified Thomas test they evaluated the dominant lower limb. What about the Isokinetic Muscle Strength Test? Was it done bilaterally or was it just for the dominant lower limb? Please specify.
- Authors stated that these tests have a good inter-rater reliability. Please report the level and the related reference.
- It is not clear how the sample recruitment phase took place. The study should be replicable. Please specify.

Validity of the findings

- Line 212-214 Please don’t report the level of non-significant. Just report “p>0.05”.
- At the beginning of the discussion, it should be reported what the purpose of the study was and whether the hypothesis has been confirmed or refuted.

Additional comments

- No comment.

·

Basic reporting

no comment

Experimental design

no comment

Validity of the findings

no comment

Additional comments

General comments:

This study investigates the differences in muscle strength and flexibility of the hip joint of professional soccer players based on their field position.

For several paragraphs, there are vague references. That would be great if the authors added more references to support the researchers’ statement.

Specific comments:

Line 58-60: “It requires frequent changes in the movement…that is interment repetitive in nature.” Is this sentence a researchers’ statement? May authors add references to support it

Linen 67-68: “Midfielders cover around 5-15%...with high intensity than attackers.” This sentence is hard to understand.

Line 71-72: “Muscle strength and flexibility have a significant role in soccer performance. Isokinetic peak torque has been used to assess muscle strength in the lower limbs among soccer players.” May authors add references to support it

Line 160-170: Authors may provide the images to describe how the test (procedures) was done. That may make it easier for the readers.

Line 171-173: For the isokinetic strength testing, the participants performed ten minutes warm-up with a cycle ergometer without resistance. The participants performed the maximal contraction through the range of motion (ROM). Why did the authors choose the warming up for 10 minutes?

Line 177: check the typo

Line 227-229: “This may have been influenced by player training designed for technical and tactical movements (Costa Silva et al. 2015) and the different tasks performed in the field (Gorostiaga et al. 2009).” Are these studies support your result? If so, in my opinion, athletes may have different strengths when they have different tasks and training. So, the studies may be opposite to your results.

Line 239: check the typo

Line 234-242: The reasons and supporting references didn’t support answer your result

Line 250-253: what kind of specific training? The authors may explain more “why” the goalkeeper has different flexibility than another player.

Reviewer 3 ·

Basic reporting

Dear authors, I think your article is very interesting and well written. References are adequate and the work fits into the field of knowledge. However, I think some corrections should improve the manuscript “Determining the hip joint isokinetic muscle strength and range of motion of professional soccer players based on their field position”.
Generally, your hypothesis is clear, and your introduction is smooth to read. I would advise you to do not to report the name of all authors you cite in the introduction text and to use “a study, one research, some authors, etc.”.
Also, I think the paragraph’s order could be improved by reporting the muscle strength and flexibility part (lines 71-89) near the end of the introduction. The main topic should be near the aim of your study and the hypothesis. You should proceed to generally [anthropometry and physical characteristics (https://doi.org/10.3390/biology11060823), physiological (cardiovascular, respiratory, neuromuscular, etc.), biological (aerobic, anaerobic, intermittent, etc.), and task demands (sprint, change of direction, etc.), field role characteristic] to specific (what you want to investigate).
Line 58: In my opinion “motor task” is more appropriate than “physical activities”.
Lines 59-60: can you justify this part with any reference?

Experimental design

The research question is well defined and the methods are correctly justified. Ethical standards are followed.
I have just one suggestion:
In lines 125-126 you cite Masuda et al., 2003 for methods used to compute the statistical power of their study. Did you do a priori or post hoc computation to inspect the type I error inflation? If yes, please report the results. If not, please compute it.

Validity of the findings

All underlined data have been provided and the results are well described. However, I would advise you should not report in the text the same numbers shown in the tables (p=.008, p=.043, etc.). You should report only numbers that are not presented in the table/figure and describe the main results of your analysis. The same information is reported twice

---

## Round 0.2 · accepted · Accept

Dear Authors,

Your manuscript is much improved. It is now worthy of publication in PeerJ. Congratulations!